# Two Years of Viral Metagenomics in a Tertiary Diagnostics Unit: Evaluation of the First 105 Cases

**DOI:** 10.3390/genes10090661

**Published:** 2019-08-29

**Authors:** Verena Kufner, Andreas Plate, Stefan Schmutz, Dominique L. Braun, Huldrych F. Günthard, Riccarda Capaul, Andrea Zbinden, Nicolas J. Mueller, Alexandra Trkola, Michael Huber

**Affiliations:** 1Institute of Medical Virology, University of Zurich, 8057 Zurich, Switzerland; 2Division of Infectious Diseases and Hospital Epidemiology, University Hospital Zurich, 8091 Zurich, Switzerland

**Keywords:** metagenomic sequencing, virus infection, virus, clinical impact, diagnostics

## Abstract

Metagenomic next-generation sequencing (mNGS) can capture the full spectrum of viral pathogens in a specimen and has the potential to become an all-in-one solution for virus diagnostics. To date, clinical application is still in an early phase and limitations remain. Here, we evaluated the impact of viral mNGS for cases analyzed over two years in a tertiary diagnostics unit. High throughput mNGS was performed upon request by the treating clinician in cases where the etiology of infection remained unknown or the initial differential diagnosis was very broad. The results were compared to conventional routine testing regarding outcome and workload. In total, 163 specimens from 105 patients were sequenced. The main sample types were cerebrospinal fluid (34%), blood (33%) and throat swabs (10%). In the majority of the cases, viral encephalitis/meningitis or respiratory infection was suspected. In parallel, conventional virus diagnostic tests were performed (mean 18.5 individually probed targets/patients). mNGS detected viruses in 34 cases (32%). While often confirmatory, in multiple cases, the identified viruses were not included in the selected routine diagnostic tests. Two years of mNGS in a tertiary diagnostics unit demonstrated the advantages of a single, untargeted approach for comprehensive, rapid and efficient virus diagnostics, confirming the utility of mNGS in complementing current routine tests.

## 1. Introduction

Viruses are one of the major causes of human infectious diseases. The identification of the causative pathogen remains challenging in several disease settings [1,2,3,4]. Many clinical syndromes, such as viral meningitis/encephalitis, respiratory syndromes or febrile illness often call for a broad differential diagnosis. This results in extensive diagnostic testing most commonly in an iterative manner starting with the most likely pathogens. This approach is time consuming and associated with considerable costs and can, in the worst case, require weeks to deliver a final result, if at all. As rapid diagnosis is essential for initiating tailored treatment, a diagnostic test with the capacity to capture all potential pathogens in a single assessment would be a tremendous improvement [5].

Viral metagenomic next-generation sequencing (mNGS) fulfills the requirement for comprehensive testing as it enables the detection of the full spectrum of viral pathogens in nearly any clinical specimen by a single assay [6]. As a hypothesis-free, unbiased test, it offers numerous advantages compared to conventional diagnostic approaches providing the possibility to detect unexpected [7,8] or even unknown pathogens [9,10,11].

The application of mNGS as a diagnostic tool is intensely discussed [5,12,13,14,15,16,17]. Elaborate workflows for mNGS have been developed over the years [18,19,20,21] and it is increasingly successfully applied in the clinic [15,22]. As any other test, it also has drawbacks [23,24]: reference standards are often missing [25], costs are still high, assay turnover times are considerably longer than for sequence specific DNA/RNA detection by polymerase chain reaction (PCR) and sensitivity may remain lower than for specific PCR assays. These facts have to be counterbalanced once the usefulness of mNGS in correctly diagnosing diverse infectious syndromes has been formally evaluated.

In 2017, we introduced virus sequence-independent metagenomic high-throughput sequencing at our tertiary diagnostic unit and established protocols for sample preparation, sequencing and bioinformatics analysis [26]. This analysis workflow has proven highly successful enabling us to detect disease causing viral pathogens in complex clinical cases [27,28,29]. Since 2017, viral mNGS was applied in 105 primarily hard-to-diagnose cases upon request by the treating clinician. In this study, we systematically evaluated the impact of mNGS on diagnosis, comparing outcome and workload to results obtained by conventional routine testing in these cases.

## 2. Materials and Methods

### 2.1. Ethical Statement

Samples were obtained in the frame of the ongoing diagnostic workup from patients of the University Hospital Zurich and other regional hospitals. The overall analysis was performed on all anonymized samples obtained. For a more detailed clinical analysis, written informed general consent of the University Hospital Zurich was sought and obtained from 67 participants, referred to as the study subgroup.

### 2.2. Study Design

In this study, all mNGS assays performed at the Institute of Medical Virology on clinical cases analyzed over a period of two years (May 2017–June 2019) were evaluated and compared to conventional routine testing regarding workload and outcome. Criteria for performing metagenomic analysis were either: (i) an unknown etiology of infection even after extensive conventional testing; or (ii) a very broadly formulated initial differential diagnosis. The analysis was either requested by the treating physician or the infectious disease (ID) consultant service of the University Hospital Zurich.

### 2.3. Viral Metagenomic Sequencing of Clinical Samples

Clinical samples were pre-processed upon arrival and nucleic acid extracts were collected and batched for weekly sequencing runs. mNGS and data analysis were performed as described previously in a subset of patients (*n* = 11, protocol v1.0.0, [26,28]). The remaining patients (*n* = 94) were analyzed using an adapted metagenomic workflow, which was established during the past years (https://github.com/medvir/virome-protocols). Detailed description is provided in Appendix B. Briefly, samples were centrifuged and filtered (0.45 µm). Total nucleic acids were extracted followed by reverse transcription with random hexamers and second strand synthesis in separate reactions for RNA and DNA genomes. Next, sequencing libraries were constructed using the NexteraXT protocol (Illumina, San Diego, CA, USA) and sequenced on a MiSeq for 1 × 151 cycles using version 3 chemistry. A maximum of five samples (plus a negative control) were sequenced per run, resulting in on average 6.8 million reads per sample. Reads were analyzed with a dedicated bioinformatic pipeline called “VirMet” (github.com/medvir/VirMet/releases/tag/v1.1.1) [27]. The raw viral sequencing reads have been uploaded to zenodo (doi: 10.5281/zenodo.3355228).

### 2.4. Criteria for Positive Virus Hits

Sequencing was considered positive for a specific virus species if: (i) at least three reads of that species were detected; (ii) the reads were distributed over the whole genome with a high coverage score (coverage score = number of bases covered/number of bases expected to be covered with the amount of virus reads); (iii) the virus was not detected more than 100 times more often in the negative control or in other samples of the same run (carry-over, index hopping); and (iv) the virus reads were detected in the corresponding workflow (RNA/DNA). While for research purposes all detected viruses were listed (“virus found” in Appendix A), only viruses that met all defined detection criteria and those with a potential clinical significance for humans were reported to the physician (“virus reported” in Appendix A; e.g., Brome mosaic virus was clearly detected according to our criteria, but not reported). Viruses were reported without any quantitative information (e.g., read numbers).

### 2.5. Evaluation of the Utility of Viral Metagenomic Sequencing Compared to Conventional Testing

The utility of mNGS was evaluated by comparing outcome and workload to conventional routine testing. To evaluate the outcome, all mNGS tests (or separately by sample type) were compared to the respective conventional test (same sample type, direct detection method, time point ±2 days apart) and positive and negative percent agreement (PPA/NPA) were calculated. Overall percent agreement (OPA) was additionally calculated as a direct comparison metric. For these analyses, co-infections and multiple tested specimens per patient were treated as individual datasets.

The evaluation of the workload for conventional testing was done in two ways. As a first measure, all viral targets were considered individually and a mean of performed tests over all patients was calculated. As a second measure, viral targets were grouped according to multiplex panels (ePlex Respiratory Pathogen Panel, GeneXpert Flu/RSV, Fast-Track Diagnostics (FTD) Respiratory pathogens 21, RIDAGENE Gastrointestinal panel). The number of panels and additional individually performed tests was named “diagnostic requests” and its mean over all patients was calculated. In both ways, all virus detection methods were considered, including specific PCR, immunofluorescence, serology, culture and intrathecal antibody synthesis testing.

### 2.6. Evaluation of the Clinical Impact

A subgroup of patients (*n* = 67) had an extended informed consent (IC). In these patients, we retrospectively analyzed clinical charts to determine the clinical impact of the performed test.

## 3. Results

### 3.1. Case Statistics of Samples Analyzed by Viral Metagenomics

Since May 2017, 163 specimens from 105 patients (one case per patient) were analyzed by mNGS with increasing numbers over the years (2017: 11 patients; 2018: 53 patients; 2019 until 12 June: 41 patients). In some cases, multiple specimens were sequenced. The most frequently analyzed sample type was cerebrospinal fluid (CSF) (34.4%), followed by blood samples (including ethylenediaminetetraacetic acid (EDTA) and native blood, 32.5%), throat swabs (9.8%), stool samples and biopsies (5.5%) (Figure 1A). Diverse departments and units requested mNGS. The test was most frequently ordered by the subspecialties neurology (30.7%), general internal medicine (including intensive care units) (19.6%), infectious diseases (11.7%) and cardiology (6.7%) as well as the emergency department (8%) (Figure 1B). CSF was the major sample type analyzed for the neurology department (56%, 28 specimens), but also blood samples and brain biopsies were processed (30% and 10%, 15 and 5 specimens, respectively). General internal medicine and infectious diseases primarily sent blood samples for analysis (37.5% and 57.9%, 12 and 11 specimens, respectively). For pulmonary diseases, either throat swabs (66.7%, 4 specimens) or broncho-alveolar lavages (BAL; 33.3%, 2 specimens) were analyzed. The remaining clinics sent in more diverse specimens, including punctures (pericardium, pleura), stool, urine, and swabs from different locations. Some rare analyzed specimens included a DNA extract of a myocardial biopsy prepared by the ordering center, a sonicate of an infected knee implant and cell culture supernatant (Figure 1B).

### 3.2. Viruses Detected by Viral Metagenomic Sequencing

Viral mNGS was performed for all 105 cases as requested by the treating physician. In 43 cases (41%), mNGS found at least one virus. Among these, mNGS detected multiple viruses in 13 cases (30.2%) resulting in a mean of 1.6 detected viruses over all positive cases. In one case, four viruses were detected in one sample (Patient 39, Appendix A). In all others, no relevant viral reads could be detected (excluding bacteriophages, plant viruses and endogenous retrovirus sequences).

In the 43 positive cases, mNGS detected 32 distinct virus species belonging to 14 virus families. Anelloviruses, which are considered body flora [30], were found most frequently (21%), followed by Herpes- and Flaviviruses (18% and 15%, respectively, Appendix A). Of those, 27 species and 13 families were reported to the physicians (Figure 2). Here, we included one case, where the detection of bacteriophage Aeromonas virus phiO18P supported microbiological routine workup which was known to be positive for *Aeromonas*.

Overall, more DNA than RNA viruses were found (17 DNA and 15 RNA virus species, Appendix A), whereas less DNA than RNA viruses were reported (13 DNA and 14 RNA virus species, Figure 2). An interactive version of the krona charts (Figure 2 and Appendix A) including found and reported viruses is provided in Appendix A.

### 3.3. Outcome of Viral Metagenomic Assay Versus Conventional Testing

We analyzed individual results of mNGS compared to routine diagnostic testing. In cases where mNGS yielded a positive result, often no respective conventional test (same viral target/direct detection method/similar time point) was done for specimens analyzed by metagenomics. Therefore, these samples were excluded for the following analyses. For the remaining 75 datasets, positive percent agreement (PPA, sensitivity) was 65% and negative percent agreement (NPA, specificity) was 95%. Twelve samples tested positive by specific PCR but not by mNGS; 10 of these tested low positive and are probably out of the detection range of mNGS. Excluding these, the sensitivity would be 92% (Table 1). The low positive conventional tests missed by mNGS were different species of Human herpesviruses and one case of Dengue virus. The two positive conventional tests detected Human gammaherpesvirus 4 (EBV) in BAL and CSF, respectively (Patients 8 and 96, Table 1).

To disentangle differences in the outcome of mNGS compared to conventional testing regarding different sample types, we performed an additional analysis for the three most frequently analyzed specimen types. Including tests reported as low positive by respective conventional testing, overall percent agreement (OPA) is 81% in CSF, 68% in blood samples and 100% in the tested throat swabs (Table 1).

We next looked at the cases with a result reported positive by mNGS that were excluded from the analysis above because no respective conventional test was performed. If we excluded viruses that are considered body flora (Anelloviruses [30]) or common skin contamination (Papillomaviruses [21,32]), mNGS detected 24 “infections” of 11 different virus species which were not tested for by a respective conventional test: Pegivirus C (7), Human betaherpesvirus 7 (4), Norwalk virus (3), Human immunodeficiency virus 1 (2), Hepatitis B virus (2), Influenza A virus (1), Human alphaherpesvirus 1 (1), Human alphaherpesvirus 2 (1), Hepatitis C virus (1), Betacoronavirus 1 (1), and Aeromonas virus phiO18P (1).

### 3.4. Workload of Viral Metagenomic Assay Versus Conventional Testing

mNGS is an open and unbiased approach with the potential to replace multiple specific diagnostic tests. We evaluated the number and workload of conventional tests that were performed in the cases analyzed in this study. We distinguished between: (i) the individually performed tests; and (ii) “diagnostic requests”, which included syndromic panels (multiplex assays, block analyses) and additional individually performed tests. For example, a single request for a multiplex panel can target multiple different virus species or subtypes. The rationale behind the two categories was the simplified differential diagnostic algorithm and the lower workload (hands-on time) when using multiplex panels. In our 105 cases, the physicians placed on average 10.8 diagnostic requests that covered 18.5 different viral targets, which shows the inherent difficulty in obtaining a correct and rapid diagnosis in these patients and the large number of viruses taken into consideration.

On the other hand, mNGS was performed once per patient, except in seven cases where an additional test was ordered at a later timepoint always including a different specimen. 70% of these additional metagenomic tests were performed on biopsies. Overall, this resulted in a mean of 1.07 mNGS analyses over all patients.

### 3.5. Timing Viral Metagenomic Assay Versus Conventional Testing

We were further interested in temporal aspects of mNGS in clinical use. We looked at the delay until mNGS was considered by the treating physician or ID consult service. Most often, mNGS was requested the same day as the sample was taken, i.e., ID specialist recognized the cases as hard-to-diagnose and necessitating an open diagnostics approach (Figure 3A). In other cases, it took several days until the test was requested, probably because in the meantime routine diagnostics did not provide a positive result. Tests requested after a prolonged time period correspond to retrospective analyses of archived specimens. In the majority of cases (65.4%), the results were validated and reported to the clinicians within seven days (Figure 3B). For urgent cases and with optimal timing (sample available early in the morning), the results could even be provided the next day. In some cases, however, it took several weeks until a final result was reported, due to delays in sample shipment or batching of less urgent cases into one sequencing run.

### 3.6. Patient Characteristics in the Study Subgroup

For 67 patients, an extended collection of clinical data was available. Basic demographic and clinical characteristics are shown in Table 2: 43 (64.2%) patients were male and the median age was 53 years. In total, 24 patients (35.8%) were immunocompromised. Most of the tests were ordered for patients treated at the department of neurology (*n* = 26 (38.8%)), and the department of general internal medicine (*n* = 15 (22.4%)). ID consultant service was involved in 45 (67.2%) cases. CSF, blood and throat swabs were the most common clinical specimens (Table 3). mNGS was most often performed in patients with a suspected meningitis/meningoencephalitis, peri-/myocarditis and febrile syndromes (Table 4).

### 3.7. Clinical Impact of Viral Metagenomics Results in the Study Subgroup

While specific treatment is often not available against viruses, a rapid confirmation or exclusion of a viral infection is important to limit nosocomial spread and avoid unnecessary antibiotic treatment. Reviewing patient charts in the study subgroup, we were interested if the positive or negative result of mNGS had a direct impact on patient treatment on top of routine diagnostics. In three cases, mNGS provided the final diagnosis for viruses not considered or available by routine diagnostic tests (see Case Vignettes 1–3 in Appendix C). Further diagnostic tests were therefore not necessary and empirical antibiotic treatment could be stopped [29]. In other cases, mNGS essentially contributed to exclude an actively replicating virus infection. In one heart transplant recipient, treatment was adjusted after confirmation of a positive routine test (Norwalk virus PCR) and exclusion of further viruses in a stool sample. One patient with an inflammatory central nervous system disorder was treated with high dose immunosuppression after exclusion of a viral infection by mNGS.

## 4. Discussion

In the present study, we evaluated the clinical utility of mNGS by assessing all cases (*n* = 105) submitted for mNGS to our tertiary diagnostics unit since May 2017. The 105 cases comprise a study set with highly diverse clinical syndromes. All were referred for mNGS by the treating physician or the ID consult service because prior routine testing had failed to define an etiology of infection or because the initial differential diagnosis was very broad. Clinical chart review revealed that, in three cases, mNGS detected a virus infection that fitted the observed clinical picture but was not considered by the original differential diagnosis and hence initially not tested by routine diagnostics. mNGS thus provided the final diagnosis in these cases and proved to be a helpful or even decisive diagnostic tool with a relevant impact on treatment decisions and case management. As determined in the outcome analysis, for 22 datasets (29.3%), mNGS matched with routine diagnostic results. In most of the samples (52%), neither mNGS nor routine diagnostic testing could detect a viral pathogen. In such cases, it remained open whether a non-viral pathogen was the cause or whether no pathogen was involved at all. Negative results are nevertheless important for antibiotic treatment decisions and lifting of precautionary isolation measures. However, it is not possible to rule out viral infections entirely as sampling may have been post peak virus production and remaining virus loads may have been below the sensitivity threshold of mNGS.

Given the large number of different viruses and virus families detected in the positive cases, it might be difficult to assign whether a detected virus is really relevant for the observed symptoms, in particular for viruses not commonly detected in humans. On the other hand, this is an advantage of the unbiased approach as potentially clinically relevant pathogens might not be considered by a conventional approach. For reporting test results to clinicians, we decided to report positive virus hits as binary results without quantitative information as for any other qualitative specific test.

Our current analysis strategy focused solely on viral pathogens. As parallel assessment of bacterial or fungal pathogens by metagenomic sequencing would provide a comprehensive analysis of infectious agents, we are currently adjusting our procedures to implement this.

On average, in the analyzed cases, physicians requested 10.8 diagnostic tests which covered 18.5 different viral targets. This highlights the inherent difficulty in obtaining a correct and rapid diagnosis in these patients and the large number of viruses taken into consideration. A single, unbiased metagenomic assay could not only limit the workload for diagnostic labs, but also be overall more cost effective, less work intensive and faster, given the high number of individual diagnostic requests on average over all patients.

The amount of nucleic acid background in clinical samples will influence the performance of metagenomics, which together with the variable sensitivity of mNGS for different virus classes and strains noted in the literature [33], renders the establishment of a reference standard very challenging. A direct comparison between standard diagnostics and metagenomics always depends on the definition of the gold standard, adding to the difficulty in validation of such a new method. Internal validation by specific PCR is key to determine sensitivity and specificity. As a quality control assessment, inter-laboratory ring trials are necessary to test any new metagenomics pipeline, to disentangle sources of variability, to formulate best practices and to gain the trust of the clinician for successful application in the clinical setting.

Despite the indisputable innovation and potential, the exact role for metagenomics in the diagnostic algorithm still has to be defined. In our study, mNGS was often ordered as a first-in-line diagnostic tool, in parallel with routine diagnostic tests. As mNGS is currently used for difficult-to-diagnose cases, there is a strong bias towards patients where no diagnosis can be made—which should not be held against metagenomics but rather seen in a clinical context, where non-infectious causes may contribute to some of the clinical syndromes. The lower sensitivity of mNGS compared to specific PCR will, at least in the initial phase, make it necessary to do specific tests in parallel, importantly for pathogens where a rapid treatment initiation and diagnosis is crucial. Given that often more than one virus is detected, a very close interdisciplinary collaboration with the clinician is key to put the findings into a clinical context.

## Figures and Tables

**Figure 1 genes-10-00661-f001:**
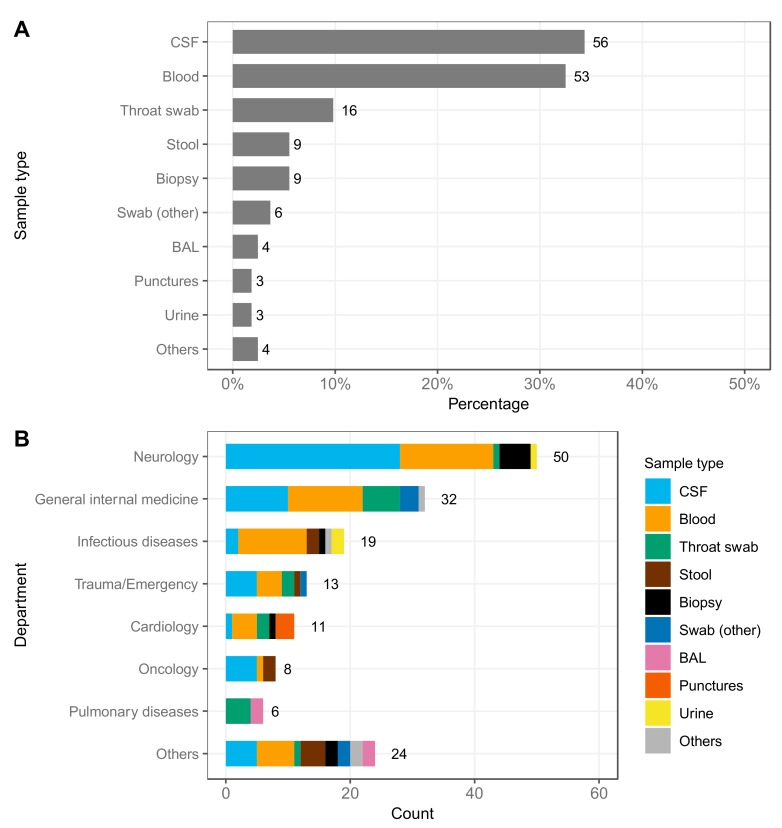
Case statistics of samples analyzed by viral metagenomic next-generation sequencing (mNGS). (**A**) Sample types analyzed in 105 cases over two years shown in percentage. Swab (other) includes enoral swab, skin swab, nasopharyngeal secretion and unknown swab. Others includes DNA extract of myocardial biopsy, sonicate of a knee implant and cell culture supernatant. (**B**) Number of samples and sample types analyzed per department. Others includes otorhinolaryngology, dermatology, rheumatology and external sources. BAL: Broncho-alveolar lavage, CSF: Cerebrospinal fluid. Bar labels indicate the total count.

**Figure 2 genes-10-00661-f002:**
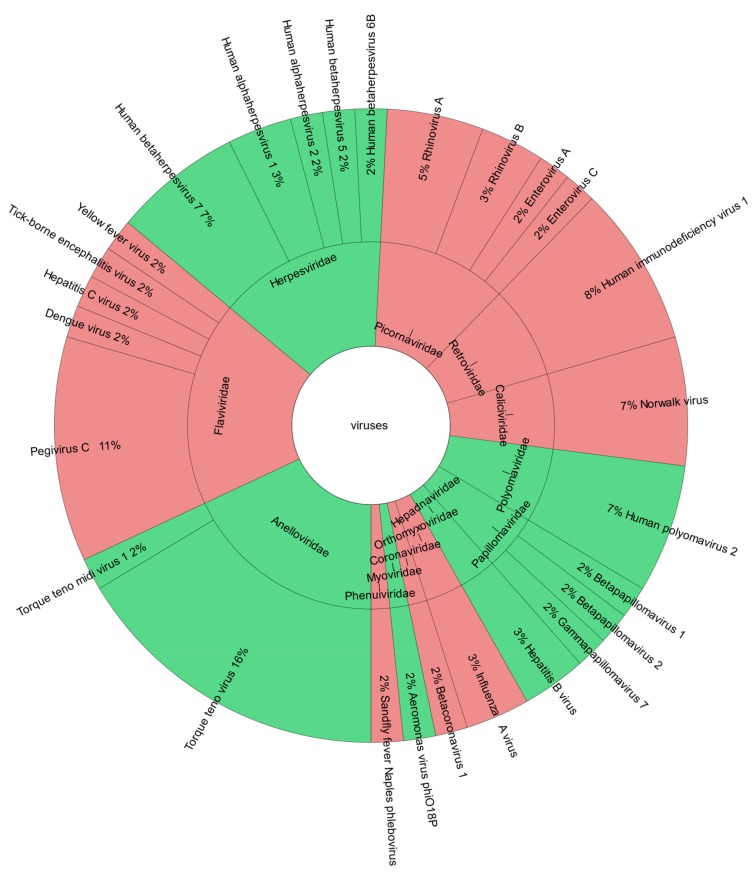
Viruses reported by viral mNGS. Krona chart of detected and reported viruses on family and species level with DNA viruses shown in green and RNA viruses shown in red. Reference krona chart: [31].

**Figure 3 genes-10-00661-f003:**
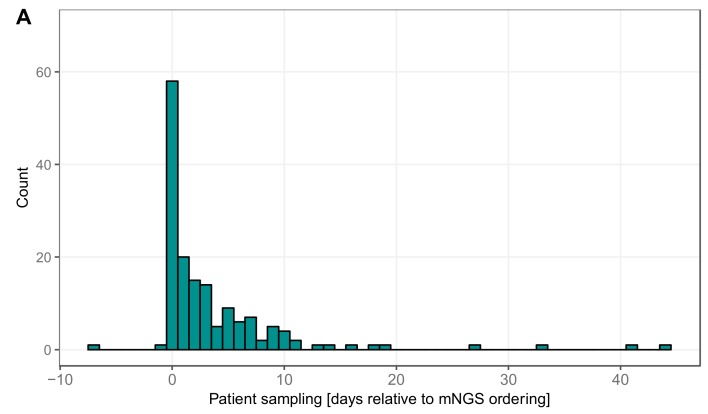
Timing of viral mNGS: (**A**) the delay (in days) between the time point when the clinical specimen was collected and the metagenomic test was requested; and (**B**) the delay (in days) between metagenomic test request and validation, corresponding to reporting of the results to the physician.

**Table 1 genes-10-00661-t001:** Comparison of the outcome of viral mNGS testing to respective conventional tests in all samples and the most frequently analyzed sample types.

					Respective Conventional Testing	
					+	–
All Samples	OPA = 81/94 ^1^	PPA = 65/92% ^1^	mNGS	+	22	2
		NPA = 95%		-	2 pos 10 low pos ^1^	39
CSF	OPA = 81/91% ^2^	PPA = 64/88% ^2^	mNGS	+	7	1
		NPA = 93%		-	1 pos 3 low pos ^2^	14
Blood	OPA = 68/100% ^3^	PPA = 46/100% ^3^	mNGS	+	5	0
		NPA = 100%		-	0 6 low pos ^3^	8
Throat swab	OPA = 91%	PPA = 100%	mNGS	+	4	1
		NPA = 86%		-	0	6

^1^ Ten out of the 12 samples that tested negative by mNGS and positive by conventional testing were actually reported as “low positive” (at the detection threshold). ^2^ Three out of the four CSF samples that tested negative by mNGS and positive by conventional testing were actually reported as “low positive” (at the detection threshold). ^3^ Six blood samples tested negative by mNGS and “low positive” by conventional testing (at the detection threshold). NPA: Negative percent agreement, OPA: Overall percent agreement, PPA: Positive percent agreement.

**Table 2 genes-10-00661-t002:** Basic characteristics in a study subgroup of 67 patients.

**Age: Median (Range)**	53 (17–88 Years)
**Male gender**	43 (64.2%)
**Patients immunocompromised**	24 (35.8%)
Post SOT	7 (29.2%)
Malignancy	5 (20.8%)
HIV	5 (20.8%)
Autoimmune disorder	7 (29.2%)
**Patients by department**	
**Internal medicine and subspecialties**	35 (52.2%)
General internal medicine	15 (22.4%)
Cardiology	7 (10.4%)
Infectious diseases	7 (10.4%)
Pulmonolgy	3 (4.5%)
Rheumatology	2 (3%)
Hematology/Oncology	1 (1.5%)
**Neurology/Neurosurgery**	28 (41.8%)
Neurology	26 (38.8%)
Neurosurgery	2 (3%)
**Other**	4 (6%)
Emergency department	1 (1.5%)
Otorhinolaryngology	1 (1.5%)
Dermatology	2 (3%)

Data shown as absolute number and percentage (in parenthesis) if not stated otherwise. HIV: Human immunodeficiency virus, SOT: Solid organ transplant.

**Table 3 genes-10-00661-t003:** Sample types analyzed in the study subgroup.

Clinical Samples:	101
CSF	35 (34.7%)
Blood	32 (31.7%)
Throat swab	11 (11%)
Biopsy	6 (6%)
Stool	4 (4%)
Urine	3 (3%)
Punctures	3 (3%)
BAL	2 (2%)
Others	5 (5%)

Data shown as absolute number and percentage (in parenthesis). Others includes nasopharyngeal swab, cell culture, skin swab, unknown swab, and nasopharyngeal secretion.

**Table 4 genes-10-00661-t004:** Tentative diagnosis when test was ordered (study subgroup).

Disease	Number of Cases
**Neurological disorders**	
Meningitis and/or encephalitis	17
Other central nervous system disorders ^1^	11
Cerebral lesion/abscess	3
Peripheral nervous system disorders	2
PML	1
**Other diseases, disorders & syndromes**	
Pericarditis and/or myocarditis	8
Febrile syndromes (including FUO)	8
Respiratory tract infections	4
Allograft dysfunction after lung transplantation	3
Diarrhea	3
Sepsis in neutropenia	1
Cytokine-Release-Syndrome	1
Unspecific polyarthritis and lymphadenopathy	1
Constitutional symptoms unknown etiology	1
Unspecific myalgia syndrome	1
Unspecific cutaneous lesions	1
Chronic sinusitis	1

^1^ Consists of patients with syndrome of transient headache and neurologic deficits with cerebrospinal fluid lymphocytosis (1), aphasia (1), multiple sclerosis (1), ocular flutter and vertigo (1), orofacial myoclonus and impairment of the oculomotor nerve (1), optic neuritis (1), cervical myelopathy (2), Parry-Romberg Syndrome (1), multiple sclerosis (1), stroke (1) and suspected cerebral vasculitis (1). PML: Progressive multifocal leukoencephalopathy, FUO: Fever unknown origin.

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
