# Peer review of "Two Years of Viral Metagenomics in a Tertiary Diagnostics Unit: Evaluation of the First 105 Cases"

_genes, 2019, doi:10.3390/genes10090661_

Round 1
Reviewer 1 Report
This is a prospective evaluation of the utility of an mNGS assay performed on patients in a tertiary medical center. This is important work since it takes up to 2 years to get 100 cases to evaluate in the first place.
There are many positive aspects of this study, including how the clinicians interacted with the ability to perform mNGS, the case vignettes, the rapid turnaround time achieved.
One major comment is that the editorial aspect of the paper is written a touch more positive than expected for mNGS missing 35% of the diagnoses because of low analytical sensitivity (possibly due to MiSeq sequencing). Rather than exclude these, it might be helpful to say what those targets and missed diagnoses were. They are spelled out for mNGS, after all.
Another major comment is that the true clinical utility of different viral infections detected. What does GB virus or HHV-7 really mean as a mNGS pickup? There's a reason they're not being tested for in the first place, unless you have new epidemiological data in their role as pathogens.
Why focus only on viruses in the context of clinical care for these patients? This is a major limitation for this mNGS manifestation.
Finally, it is not clear that the authors could have been validated for all the specimen types tested. It would help to know more about the validation/performance of the assay in the different specimen types and I encourage the authors to make a Figure and additional Results section on that aspect.
line 254 - "proved" instead of "proofed"
line 175-178 - why is the phlebovirus not in this list?
Figure 2 - font formatting issues on 5-6 of the entries.
Figure 3 - what is the story with the samples tested after 50 days?
line 281 - I find the suggestion of ring trials to be a bit odd in this scenario.
Reviewer 2 Report
The manuscript by Kufner et al. describes an effort to establish a practical metagenomics sequencing system for diagnosis of human cases of unknown etiology. By performing viral metagenomic next-generation sequencing analysis through the VirMet pipeline with clinical specimens of cerebrospinal fluid, blood, and throat swabs collected from 105 patients, the authors succeeded in detecting various species of viruses, both DNA and RNA viruses, and found that some of the detected viruses were not included in conventional testing. Furthermore, they presented several examples in which the results from NGS helped determine the medical management of the corresponding cases.
Overall, the manuscript is well-written, and provides a valuable set of data for viral metagenomic sequencing designed for clinical use.
Several minor issues should be addressed before publication.
Page 5, line 149: “Overall, more DNA than RNA viruses were found (15 DNA and 17 RNA virus species, Suppl Figure S1)” Does this mean more RNA viruses than DNA viruses? Regarding the NGS procedure, how many samples were simultaneously analyzed in the same run? What is the average read number per each sample? Is there any trend that RNA viruses are difficult to detect when the delay is prolonged relative to NGS ordering? Page 8, line 214: It is wired that the median age is “51.8 years”. The median should be the number without decimals because the patient number is odd. Also, the percentage shown in Table 2 seems to be incorrect. Please check it again. ID does not appear in Table 2.
Reviewer 3 Report
Kufner et al. performed viral mNGS testing from 105 patients with suspected infection. They found good concordance with standard clinical testing, identified viruses that would not have otherwise been detected in 3(?) patients, and found that mNGS testing was timely and impactful for patient care in several cases. Overall, this is a strong study that demonstrates the utility of mNGS across a range of clinical syndromes and adds valuable information to this exciting and rapidly-evolving field. The paper would benefit from some additional detail and clarification.
# In the Results section (and Table S1), it seems that there were 25 viruses detected by mNGS that were not tested by a conventional test (lines 174-178), however in the Discussion section, only 3 cases were reported as being detected by mNGS but not tested by routine diagnostics (lines 251-254). Please clarify.
# Readers would benefit from additional discussion of how mNGS results were reported to clinicians, whether considered a research test vs. clinical diagnostic test, and how it was determined whether the detected viruses were pathogens. For example, in Table S1, how was it determined that the herpesvirus 4 and 5 found in CSF were not pathogens?
# Were the viruses that were detected (Figure 2) found in specimen types and clinical scenarios that suggested they were true pathogens? Why was there aeromonas phage but not other phages included in this figure? Some of this information can be gleaned from Table S1 but it would help the reader to interpret this more explicitly in the text.
# How deeply were the samples sequenced? It seems plausible that deeper sequencing might allow the recovery of viral reads in some of the samples that were negative by mNGS and “low positive” by conventional testing (Table 1). Because these samples were indeed positive by conventional testing, I would not report the sensitivity of mNGS excluding them (line 165).
# Were there viruses commonly detected in the negative controls and if so what were they? Please clarify on line 90 “the virus (if itself present at a low number of reads) was not detected in the negative control” – what constitutes a low number of reads?
# Which two viruses were missed by mNGS but detected by conventional testing (Table 1) and was the conventional test PCR or non-nucleic-acid-based?
# Please confirm consistent use of terminology throughout the paper; e.g. sometimes “cases” seems to refer to patients (line 247) while other times it refers to samples (line 255), and this makes it challenging to keep track of the numbers.
# Please clarify “ring trials” (line 281)
# I don’t know that it is commonly accepted that pegivirus C causes encephalitis, even as written in the series cited by the authors (Vignette 3); would suggest interpreting this more conservatively. Were there RBCs detected in the CSF sample? Is it possible that pegivirus from blood was introduced during the lumbar puncture?
